# "COVID-19 was a FIFA conspiracy #curropt": An Investigation into the Viral Spread of COVID-19 Misinformation

## Abstract

The outbreak of the infectious and fatal disease COVID-19 has revealed that pandemics assail public health in two waves: first, from the contagion itself and second, from plagues of suspicion and stigma. Now, we have in our hands and on our phones an outbreak of moral controversy. Modern dependency on social media has not only facilitated access to the locations of vaccine clinics and testing sites but also—and more frequently—to the convoluted explanations of how "COVID-19 was a FIFA conspiracy" [1]. The MIT Media Lab finds that false news "diffuses significantly farther, faster, deeper, and more broadly than truth, in all categories of information, and by an order of magnitude" [2]. The question is, how does the spread of misinformation interact with a physical epidemic disease? In this paper, we estimate the extent to which misinformation has influenced the course of the COVID-19 pandemic using natural language processing models and provide a strategy to combat social media posts that are likely to cause widespread harm.

## 1 Introduction

Numerous technology companies have already implemented machine learning algorithms to obstruct the spread of false information. Instagram and YouTube have both pledged to curb the amount of deceitful posts "that pose a serious risk of egregious harm" on their platforms while not inhibiting the freedom of expression of their users through False Information [3] and Intelligence Desk [4] respectively. With the prevalence of misinformation in the media, it is of the utmost importance to limit the reach of false authoritative content regarding the COVID-19 pandemic, especially when their main victims are regular civilians. Our research has culminated in a misinformation detection pipeline that is comprised of three components: a claim detector, a misinformation classifier, and a virality measurement. Through this pipeline, we aim to derive further insights into the behavior of these types of information spread and their impact on society.

## 2 Related Work

### 2.1 ClaimBuster

Full Fact has created real-time automated fact checking tools that first identify and label each sentence according to the type of claim it contains (e.g. claims about quantities, claims about cause and effect, and predictive claims), then check if the given input matches something previously fact checked. We have decided to operate under their working definition of a claim: sentences where the public would

want to know its truthfulness [5]. We took inspiration from its active classification system which compares a sentence's information with data from the UK Office for National Statistics API [6]. The current state-of-the-art benchmark is ClaimBuster, which contains a monitor for text retrieval, a spotter for identifying verifiable claims, a matcher for finding existing fact-checks to the claims, a checker for querying external sources when a fact-check is not found, and a reporter that reports results from the matcher and checker to the public. The classification model incorporates TF-IDF, part-of-speech tags, and named entity recognition features and produces a binary score representing whether a claim is checkable or not. The claim spotter models were trained on a dataset of U.S. general election presidential debates labeled as Non-Factual Sentences (NFS), Unimportant Factual Design Sentences (UFS), or Check-worthy Factual Sentences (CFS) [7].

## 2.2 Tweet Legitimacy Classifier

The classification of social media content as legitimate or misinformation falls under the task of fake news detection. As both require an efficient solution to measure a statement's truthfulness, linguistic-based methods tend to outperform purely network-based approaches that assess the source's credibility. These linguistic-based methods instead contend with a statement's content and find patterns within the text that characterize that of fake news. BERT is one such state-of-the-art transformer-based machine learning model that is frequently used in language modeling. Models that are pre-trained on general, non-professional corpuses such as Wikipedia can achieve 98% and 99% precision, 99% and 97% recall, and 98% and 98% F-1 score for real and fake news respectively [8]. Due to this stage of unsupervised pre-processing, BERT models form an integral part of language understanding systems by reducing the need to build "heavily-engineered task-specific architectures" [9].

## 2.3 Virality Analysis

Research on the virality of Tweets has largely centered on retweets. A study on COVID-19 related Tweets shows that celebrities' Tweets outperformed those by health and scientific institutions, which is in line with the intuition that factors such as overall outreach beyond the Tweet's content have a tremendous impact on the spread of a Tweet [10]. Specifically, the most important features for predicting the number of retweets are the number of followers, as well as the usage of URLs and hashtags, all of which have a positive correlation with the number of retweets [11]. Another such factor is that someone who posts more statuses is more likely to be retweeted [12].

## 2.4 Sources of Data

The CMU-MisCov19 [13] dataset contains about 4,600 Tweet IDs relating to COVID-19 claims. These Tweets were hand-labeled into 17 categories representing various aspects of COVID-19 misinformation, such as true treatment, true prevention, conspiracy, fake cure, fake treatment, false fact, politics, and panic buying. Another dataset is procured out of USC [14] which contains an exhaustive quantity of ~2.2 billion Tweets pertaining to anything related to COVID-19.

# 3 ClaimBuster

## 3.1 Setting

To filter non-claim based statements, we utilize ClaimBuster [15]. This claim detection model acts as the gatekeeper of the pipeline to ensure that the assumptions in CMU-MisCov19 hold true in a natural setting. We use USC's [14] dataset for basic verification checks in Section 6.

## 3.2 Experiment

The first step of the pipeline is to distinguish claim-based data from their counterpart. We adopted the ClaimSpotter model from the ClaimBuster architecture to assign a label to each Tweet in our data as a transfer learning process.

Given three options, bidirectional LSTM, SVM, or adversarial training on transformer networks [16], we settled on the bidirectional LSTM as it offered the most configurability and consistently outperformed the other models. Though the SVM model is significantly faster to train, the model is too simple to capture the complexity of Tweets' syntactic and semantic features, and even using a Gaussian kernel did not lead to convergence. Meanwhile, the adversarial transformer networks were too slow to fine-tune. Adopting a smaller bi-LSTM architecture would be a more efficient choice, which is capable of utilizing both past and future contexts.

In order to apply their existing model to our sample data, we structured our data in the same format as that of the original ClaimBuster model. However, that original input consisted of a single sentence, whereas each sample Tweet may contain multiple sentences. A solution would be to parse Tweets into smaller chunks than sentences. However, a great portion of Tweets produce unreasonable results as they are too short or express strong support for another (unmentioned) Tweet. Moreover, one Tweet may consist of both claims and non-claims. Separating a single Tweet to these two parts produces irrelevant information as the main purpose of this step is to remove nonsensical and non-claim Tweets from the dataset. Producing more than one prediction on one sample data would obscure the task. Hence, we apply the ClaimSpotter model on one Tweet as a whole.

The ClaimSpotter model has been reported to achieve 0.74 in recall and 0.79 for precision [15] under the context of analyzing presidential debates. Although the context is significantly different, we believe that unlike content and types of language used, claims as a linguistic component should be universally transferable, thus re-training on a Twitter specific dataset was not performed (not to mention the difficulty of hand labelling a large enough dataset). Further ClaimSpotter verification results can be found under Section 6.

## 4    Tweet Legitimacy Classification

We constructed training and validation datasets from CMU-MisCov19 [13], which to train our multi-class Tweet legitimacy classifier, we binned these 17 themes into legitimate, misinformation, and irrelevant information in the context of COVID-19.

As a statement's truthfulness can affect its reach, we first developed a model to identify real, fake, or irrelevant to COVID-19 information. With the given dataset of Tweets representing social media posts in general, we adapted existing natural language processing techniques to this specific task and input. Specifically, we fine-tuned Digital Epidemiology's COVID-19 specific BERT model—Covid-Twitter-Bert-v2—on [17] the binned CMU-MisCov19 dataset.

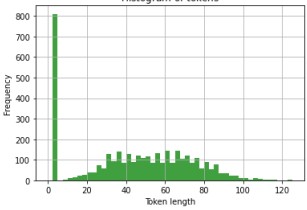

Figure 1: Histogram of token lengths

First, we determined the maximum token length for our inputs. Since this hyperparameter greatly affects training time and memory usage, it should be delicately selected. We performed a CDF calculation (using figure 1) and found that >90% of our data was less than 96 tokens. Hence, we chose a maximum token length of 96.

Further, we employed text preprocessing techniques on the Tweets to reduce the amount of time the model took to converge. Each Tweet was parsed as raw text and fed into the following pipeline: 1) make lowercase, 2) remove URLs, 3) remove mentions of other users, 4) remove non-ascii characters, 5) remove punctuation, 6) remove stop words (using NLTK's stopword bank), 7) lemmatize the words

to their root words using the NLTK library. This technique reduced the training wall-clock time by about 3x on our hardware, which made the rest of this experiment feasible. The fine-tuned model achieved about ~74% accuracy on the dataset with no further modifications.

To improve our classifier's accuracy, we increased our training dataset and implemented an ensemble model through bagging. We augmented the number of datapoints by hand-labelling a random sample of 2,005 tweets from the USC dataset according to the original MisCov19 methods. Then, we trained the same BERT model on that augmented dataset and achieved accuracies up to 79%. Table 1 summarizes our model accuracy and loss on the validation set.

Table 1: Tweet Legitimacy Classification Model accuracy and loss of single model

| Model | Validation Loss | Validation Accuracy |
| --- | --- | --- |
| Fine-tuned on original MisCov19 Dataset | 0.6446 | 0.7447 |
| Fine-tuned on augmented MisCov19 Dataset | 0.8008 | 0.7910 |

Lastly, we created an ensemble model by combining four BERT models trained on the augmented dataset. We utilized a bagging method by extracting the probabilities each model assigned to each label for a given input and averaging them. This took into account how "confident" a model was in labelling any given input. Here, we use probability as a proxy for confidence. This bagging method achieved the greatest accuracy: up to 84% on the original dataset and 87% on the augmented dataset.

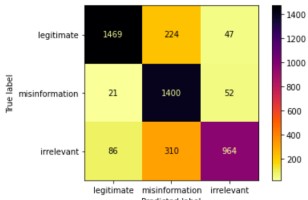
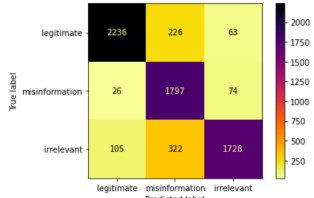
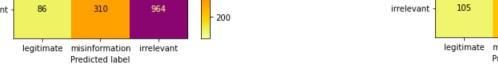

Figure 2: Confusion matrices on original vs. augmented MisCov19 dataset

In summary, the greatest challenge of achieving high accuracy on Tweet misinformation detection is input length. Tweets, by nature, are short and convey little information. Most of our misclassifications are from short Tweets that contain single misinformed facts.

## 5 Virality Analysis

### 5.1 Setting

Our data sampling method involved uniformly randomly sampling 160,000 Tweets from January 28, 2020 to December 17th, 2021 of the USC dataset [14]. This number was chosen on the basis that ~63% of Tweets were in English and were pulled from the most "active" times of the day for the platform to try and ensure more English Tweets [18]. The next step was preprocessing the Tweets' text as input into the BERT model by removing URLs, non-ASCII symbols, special symbols, and extra whitespace. We also added spaces between punctuation marks, made the text lower case, and removed Tweets of 3 or less words.

We first created a virality metric since we did not find a standardized formula in literature. Our formula was based on a Tweet's retweets, comments, and likes. However, on average, a Tweet's likes is greater than its comments and retweets. This is reflected in the training dataset as the average number of likes was 6.44, comments was 1.17, and retweets was 1.06. Thus, we normalized those features to be between 0 and 1. While retweets are the most direct measurement of a Tweet's spread,

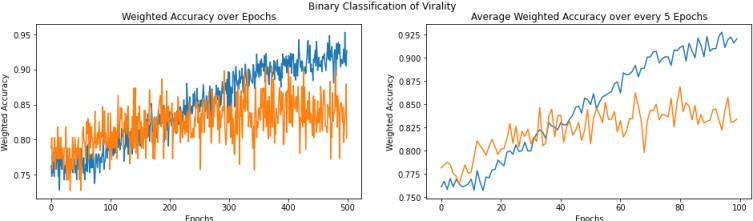

Figure 3: Virality classification performance

likes and comments remain important measures of engagement, so we decided on the equation of

$$\text{virality score} = 2(\text{retweet score}) + \text{likes score} + \text{comments score}$$

It was immediately apparent that the dataset is populated by Tweets that have very little engagement, and we will refer to Tweets having 0 likes, retweets, and comments as "dead" Tweets. However, the dataset also features some Tweets with extremely high scores. To account for this large range, we scale the virality scores logarithmically.

The next step was to determine inputs to our model. From existing literature as well as our dataset's metadata, we decided to include the Tweet text itself, number of followers, number of users they are following, number of statuses, if the user is verified, the usage of hashtags, and the usage of URLs. These features, where applicable, were also logarithmically scaled to match the virality score scaling.

## 5.2  Experiment

To predict whether a Tweet is going to be viral or not, we developed a binary classification model. The threshold for a viral claim is a score of 7.294, which, for example, corresponds to 25 retweets, 50 comments, and 100 likes. While initially this might not appear to be "viral," this score is greater than even the 99th percentile of Tweets due to the vast amounts of "dead" Tweets.

The architecture for this model consists of passing preprocessed text through the same BERT model present in Section 5 and obtaining word embedding vectors of size 1024. These are then fed through a dropout layer and two hidden layers each attached with a ReLU activation unit. The resulting output of size 26 is then concatenated with the 6 features discussed at the end of Section 6.1. Across experimentations with the ideal output size of this first head of the network's architecture, no apparent information gain is obtained after a size of roughly 26. Following this, the 32 inputs are passed through 5 hidden layers before reaching an output size of 1 and being passed through a sigmoid layer. This produces a final probability-like measurement that is rounded to obtain the class prediction.

Prior to any data resampling or distribution, a sample of 13,920 Tweets had less than 1% of its Tweets considered viral, making it difficult to not only configure a loss function that reflected such an imbalance but also directly re-sample to form more informative datasets for training. We split the training and validation dataset along an 85/15 split with mini-batch sizes of 64 while also removing 75% of the "dead" Tweets from the training dataset as well as 75% of the Tweets with virality score between 1 and 2. This presented a much more balanced—albeit unrepresentative—dataset from which we can artificially force the model to learn properly. The validation set, however, maintained a more authentic representation of the data.

For the training hyperparameters, we utilized the Adam optimizer after experiments involving other optimizers such as SGD with Nesterov Momentum and other Adam variants (RAdam and AdamW). This optimizer used a default learning rate of 0.001 which we found performed the best in conjunction with a weight decay value of 0.0005. We used BCELoss and weighted accuracy due to the nature of the task in addition to manually computing a balancing factor to weight viral Tweets as more important. It also appeared that rather than traditionally running the training loop across $X$ epochs, running $Y$ iterations of $Z$ epochs, where $Y \cdot Z = X$, performed better. Thus, the latter was used for training. A possible explanation is that resetting the learning rate after each iteration improves the

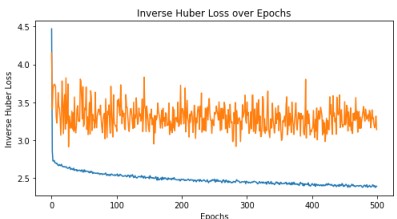

Figure 4: Virality regression loss

progress made since the adaptive learning rate is not suited to handle this complex problem. However, we found that SGD, which is not adaptive in nature, performs worse in the former training regimen. Thus, we believe that there may be some middle ground between a decaying adaptive learning rate and a constant learning rate that involves resetting the learning rate occasionally.

Figure 3 indicates that training accuracies improve substantially after 50 epochs: beginning around 76% and improves up to 92% at the 500 epoch mark. Validation behaves similarly with a starting accuracy of ~78%, which improves to ~84% within 400 epochs before decreasing. It is hard to accurately compare the two curves in the traditional sense for overfitting due to our data sampling methodology. The training loss decreases throughout whereas validation loss decreases before increasing and becomes increasingly noisier as well. Across both accuracy and loss graphs, both measurements are extremely noisy due to the sparse presence of viral Tweets and the particular randomization of Tweets in every batch.

In addition, we built a regression model to predict the degree of virality of a Tweet. This is a much more complex problem to examine due to the sparsity of viral Tweets. The model architecture remains the same except for the final sigmoid layer, or lack thereof. The data input had the same sampling scheme aside from maintaining the virality scores instead of class processing. The same hyperparameters were used as previously except the loss function was changed to an inverse huber loss function. The nature of the distribution of the virality scores skewed to the right, making the scores predicted by the regression naturally lower in value. To incentivize better learning towards the higher virality values, we maintained a constant loss for values that differ from the truth value by less than 1 and squared the loss for all that exceeded a loss of 1 to further penalize them.

The training loss continues to decrease with little noise whereas the validation loss is noisy and much higher. The higher validation loss is a byproduct of the data sampling scheme since the validation set contains lower virality score Tweets on average. This increases the average loss if its performance in that portion of the data is poor. The noise is also partially explained by the smaller quantity of data. Moreover, the model will never predict above a virality score of 6. The loss appears to plateau around a value of 2.3, which correlates with each prediction being off by ~1.5. Thus, precisely predicting the degree of virality is still a very complex and not solved problem, and a lack of an extensive dataset will also significantly hinder a model's ability to identify the characteristics of more viral Tweets.

# 6   Full Pipeline

Our full pipeline consists of the ClaimBuster, Tweet Legitimacy classifier, and the Virality analysis model. The input to this machine learning pipeline is a single Tweet, for which its text will be analyzed for truth value, within the context of the COVID-19 pandemic and its user engagement metrics will be used to quantify its impact. The practical usage involves determining whether a Tweet is claim or not, checking whether or not it is misinformation, and whether or not it is at risk of spreading to a significant audience, at which point a company, like Twitter, can make a decision to flag it. An experiment was conducted with 250 Tweets sampled uniformly and randomly across their virality scores; specifically, there are 50 Tweets for each bucket of viralness: 0-1, 1-3, 3-5, 5-7, 7+. We fed these Tweets through the pipeline to get results to yield a 78% accuracy and 0.72 F1 score for the Claimbuster model, a 84% accuracy and a 0.81 macro-F1 score for the misinformation model. It

appears that the Claimbuster confuses true claims with non-claims more often than the other way around by a significant margin. With the misinformation model, we're able to detect legitimate claims much more accurately than the other classes with a 0.91 F1 score compared to 0.77 F1 scores in both the irrelevant and misinformation classes. There is significant confusion from the model when it comes to irrelevant classes and part of that is due to the complex nature of the definition of this class category. Politics for example is defined to be part of this irrelevant category but when the context is associated with public health and government, these claims are often hard to distinguish even among humans. A trend appears to be that both models also perform worse with Tweets of mediocre virality.

Using our pipeline, we analyzed the distribution of legitimate or misinformation among claims found in Tweets of various popularity buckets. We find that the proportion of unpopular Tweets containing misinformation is 2-3$\times$ higher than that of viral Tweets. This is consistent with our hypothesis that generally people interact less with social media posts that are false or wrong. We interpret from our experiments that misinformation has been rampant. However, individual users' decisions to not interact with misinformed posts has prevented widespread disaster. We have demonstrated that our pipeline is a practical linguistics-based misinformation detection model that incorporates a Tweet's potential virality which combats misinformation.

# 7 Future Works

For the ClaimSpotter model, being able to incorporate multiple related claims into the model while simultaneously removing irrelevant phrases would significantly improve the validity of the model as it's hard to entirely classify a tweet as a claim or not since they can include a multitude of phrases. Furthermore, this can then be improved within the Legitimacy Classifier as only claim portions of the Tweet would be fed in, making for lower variance in the structure of the data.

Future work for the Tweet Legitimacy Classifier step of our project includes adapting it to longer social media posts, for which we hypothesize it will be more accurate. An additional factor that can be included to further bolster performance would be to look at historical Tweets from each particular user and include the legitimacy of those Tweets as those who post popular conspiracy theories often have a history of such behavior.

For the Virality Analysis, future work includes updating how we measure virality. One idea is to utilize the number of followers of the retweeters for a Tweet. If a Tweet's retweeters have more followers, then it is reaching more users' feeds and is a more robust measure of retweet-based spread. In addition, we could expand the analysis beyond just a single tweet's virality and look at the impact on users. For example, we could detect if a user Tweets out misinformation due to interacting with a different user's Tweet. Further improvements on the modelling side include utilizing hardware accelerators such as GPUs and TPUs to decrease runtime and allow for more complex models to be run. A possible model could include training both the pre-trained BERT model weights in addition to the weights from the standard Neural Network structure such that the word embeddings can extract more useful features from the text pertaining to virality. Google Colab Pro was used to incorporate both hardware techniques into this exact model but had insufficient memory and was thus abandoned.

# 8 Conclusion

Experiments using our full pipeline on prior data show that COVID-19 misinformation is widespread across social media, albeit less frequent in viral posts. Thus, we demonstrate the necessity for better misinformation filtering. Our pipeline serves as a practical linguistics-based misinformation warning system that is not reliant on a heavy fact-checking corpus. Furthermore, we introduce an attempt at identifying viral features of a Tweet prior to posting, opening the doors to future understanding of how misinformation propagates through the masses.

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
