# OpenReview forum: "“COVID-19 was a FIFA conspiracy #curropt”: An Investigation into the Viral Spread of COVID-19 Misinformation"
_uoft.ai/University_of_Toronto/2021/ProjectX — ProjectX2021_

### Official Review · Reviewer_RYQh · 2022-02-08
**COVID-19 was a FIFA conspiracy #curropt”: An Investigation into the Viral Spread of COVID-19 Misinformation**

**Rating:** 6
**Confidence:** 5

**Review:**

**Connection to Current Science (science and practice)**

1.5

- The related works sections 2.1-2.4 are fine. More could be said, e.g., about the model and task in Sec 2.2, about about other models of social ‘contagion’ in Sec 2.3, and about tasks applied to CMU-MisCov19 in Sec 2.4
- the images in Figures 2, 3, and 4 are unreadable without zooming in.
- while there is no single standardized formula for vitality (line ?? Between 138 and 139), there are multiple possible statistics to explore. There are multiple graph-based descriptive statistics, and the classic PageRank algorithm is sometimes used.





**Clarity of Communication**

1.5

- technically, there are other waves (line 2) apart from contagion and stigma, like systematic breakdowns (e.g., cancelled surgeries) and economic challenges (e.g., business closures)…
- generally the language is confident and correct, with good structure. Occasionally, there are awkwardly-worded phrases (e.g., " which to train” on Line 98)
- be sure to date URLs in references.



**Methodological Quality**

2.5

- What assumptions of CMU-MisCov19 are ensured by ClaimBuster (and how, line 73)?
- it is not clear how Tweets were fed into an SVM model, nor anything about that model except for the fact that one variant used a Gaussian kernel.
- Nothing is really said about the adversarial transformer networks (line 79) either.
- Why would inputting chunks smaller than a sentence be preferable to chunks longer than a sentence, if the expectation is still a sentence (line 85)?
- The ‘belief’ on line 93 that ‘claims as a linguistic component should be universally transferable’ is not explained, nor does it seem very reasonable in the context of content and the type of language being different.
- More should be said to connect the claim that “a statement’s truthfulness can affect its reach” (line 101), as that wasn’t well established in earlier sections.


**Reproducibility**

0.25

- Although detail is in the citation, more can be said around line 91 about the task and corpus used to derive those metrics.
- the ‘hand-labelling a random sample’ process on line 118 requires more detail, as does clarity around the validation set used on line 120.
- how the vitality scores are scaled ‘logarithmically’ (line 142) need precision.

---

### Official Review · Reviewer_Ttaz · 2022-02-11
**Ambitious research, lacks focus**

**Rating:** 6
**Confidence:** 3

**Review:**

### Summary of paper
The authors describe a pipeline for identifying claims, classifying tweets containing misinformation about Covid-19 and predicting how viral a tweet will be. The immediate goal is to use this trained model pipeline to understand how misinformation and false claims about Covid-19 spreads on Twitter, with the ultimate long-term goal of understanding the impact of misinformation in social media on the pandemic.

### Connection to current science & practice
Concise overview of literature. The authors could've better highlighted what's lacking in terms of prior research for this problem or related problems. Some claims were missing references (e.g. line 33, 43-44). Terms like "fake news", "[il]legitimacy" and "misinformation" were used seemingly interchangeably; they should've been defined rigorously, e.g. how would you classify tweets with misinformation written ironically, or ostensibly truthful personal experiences that clash with scientific consensus. Although the idea of classifying tweets or using BERT is not novel, I think the pipeline of claim detection - misinformation classification - virality measurement in social media/Covid-19 is interesting and novel. The pathway to implementation is specified. 2/3

### Clarity & communication
I appreciate that this study has many components and the authors have included a lot of information and done a lot of work, but the paper is a little hard to follow the first time reading and I had to re-read it a few times before I understood everything. The writing is good generally, feels like the authors could've improved the paper if they had some more time. Some links are clickable hyperlinks but others not. Plots are well-made overall but could've used a larger size font and figure legends describing what the colours mean. The abstract mentions that this study was to "estimate the extent to which misinformation has influenced the course of the COVID-19 pandemic", which is extremely broad and not really what the paper is about, so it could've been more specific. 1/2

### Methodological quality
Modelling choices are reasonably well-justified, and simplifying assumptions are further discussed in "Future Works". Some specific points:
* For tweet legitimacy classification (section 4), I commend the authors for hand-labelling additional data themselves. The ensemble method ("4 BERT models") in that section should be explained better. It seemed like what was called the "validation set" was used as a test set, not for tuning BERT models. Also the summary of this section (lines 127-128) doesn't follow from the results from that section - accuracy of tweets as a function of input length wasn't evaluated anywhere.
* The virality score seems to be a novel metric developed by the authors which they mention they did because a standardized virality score does not exist in the literature. I'd have liked some justification of their specification of the virality score function, e.g., why a `2 * retweet score`, why not `3`.
* Line 155 mentions Section 6.1, which does not exist in the paper.
* It's not obvious to me that the first step of the pipeline (i.e. claim detection) is necessary. You could've left that out and reduced the complexity of the pipeline and this research project.
* The results (lines 224-231) are interesting and potentially novel. The authors mention in the discussion that their results are "consistent with our hypothesis that
generally people interact less with social media posts that are false or wrong". Was this hypothesis based on some prior work or knowledge?
* How did you bin the 17 categories from CMU-MisCov19 to the 3 that you used for misinformation classifier, e.g. where was the original ambiguous category binned?

2/4

### Reproducibility
Seems somewhat reasonable for a 5-month project, but perhaps the authors could've limited the scope in some way for a more focused research project given the complexity of the topic. Although I wasn't able to access the github repo, the authors have mentioned that they've made their code available. 1/1

---

### Official Review · Reviewer_8Yed · 2022-02-14
**Review of Viral spread of Covid-19 misinformation**

**Rating:** 8
**Confidence:** 3

**Review:**

Connection to current science:
Background and rationale demonstrate knowledge of the current field, and application of misinformation detection in the COVID-19 context.
The abstract indicates that the work will address how “the spread of misinformation interact(s) with a. physical epidemic disease? … And the extent to which misinformation has influenced the course of the pandemic.” Neither of these are explicitly addressed in the paper. Rather, the work focuses on a way to identify and respond to twitter posts which have the potential to be false, and lead to widespread dissemination.
The authors have described how their work can be advanced through future research, and have conveyed the need for improved misinformation filtering, however they do not discuss directly how/where this work can be implemented. I would have liked to see a dedicated section between existing sections 6 & 7 that details a pathway to implementation. Some of this is briefly mentioned in section 6, but draw it out in a standalone section. Discuss the meaning of the findings/results - how can this work can be implemented? What is the impact it could have on the field? Convince your reader that it is important! Outline the strengths and limitations of the work you’ve presented. Then go on to describe the next steps / Future work.


Clarity of communication:
Beautifully written abstract. Engaging, enjoyable to read. Catches the reader’s attention and makes me want to read more.
Paper sections are clear and distinct. There is an overall logical flow to the paper.  There are a few places where the authors refer to something later in the paper, it would aid the reader to see these discussed in the present section (i.e., section. 5.2 refers the reader to section 6.1 to see the 6 features). Additional use of descriptive section headers would aid the reader.
Overall, the paper flows well and creates an engaging story written in appropriate scientific format.

Method Quality:
Where possible, the authors appear to be advancing on the work of others (Claimspotter), without unnecessary complication or duplication of work. I think the virality metric is a neat addition – but I would have liked to have seen more assessment of the validity of the metric.
What is the impact of reducing to a more balanced & unrepresentative dataset training model?
The paper would benefit from a strengths & limitations section. There are several decisions and assumptions made by the authors that will impact the results – these should be explicitly identified and described for the reader.

Reproducibility:
The work appears appropriate in complexity for a 5-month project. Significant effort was made to ensure the results are reproducible. I read through the code, which is well commented, though made no further effort to reproduce results.

---

### Official Review · Reviewer_vXkQ · 2022-02-16
**Review of “COVID-19 was a FIFA conspiracy #curropt”: An Investigation into the Viral Spread of COVID-19 Misinformation**

**Rating:** 7
**Confidence:** 3

**Review:**

Viral misinformation spread, or infodemic, or "too much information including false or misleading information in digital and physical environments during a disease outbreak" is recognized by the World Health Organization as one of the setbacks to public health. In this research paper, the team assembled a pipeline that factchecks (using ClaimBuster), classifies COVID-19 claims using a COVID-19 misinformation specific BERT model (Covid-Twitter-Bert-v2) - binning 17 themes into legitimate/misinformation/irrelevant, and detects virality (using a self-constructed metric based on retweets, comments, and likes, with a threshold score of 7.294). Using this threshold, the team set up an experiment to predict whether a tweet is viral, they  passed preprocessed text through a deep learning model (BERT) to find a probability-like measurement. The sparsity of viral tweets in the procured dataset generated difficulties in configuring loss function, amongst other problems, and subsequently a 85/15 train/validation set was established, excluding 75% of the tweets without engagement in the training set to increase model learning. Hyperparameters including the learning rate, batch size, weight decay, were tuned - noisiness was substantial in the validation set, rendering the virality prediction model unreliable.

Despite unsuccessful virality prediction, the team found that there are lots of misinformation on twitter, but they are 2-3 times more likely to remain unpopular instead of going viral, suggesting that though misinformation is rampant, people tend to interact less with false social media posts.

Several observations to the pipeline:
1. Dataset annotation: incorporate a consistent annotation schema and report the inter-rater reliability to ensure human annotations/labelling are consistent. Accuracy of the manual labels effect the predictive accuracy greatly especially when samples sizes are small or dealing with a non-binary classification.
2. The team showed care into the construction of the paper and showed both the metric loss and performance loss when describing hyperparameter tuning, it will help to describe what the blue line and orange lines represent in Figure 3 & 4.
3. The virality metric score [(2*retweet) + likes score + comments score ] was subjective but has its merits. Consider what other metadata obtained from the procured dataset can enhance the model (e.g., sentiment, use of adjectives, time of tweet, news cycles) - these are some examples that are supported by literature.

Overall it is a clean, considerate and solid work, the paper has a lot of promise as the team mentioned in future work.

---

### Official Review · Reviewer_Nc8Z · 2022-02-16
**Interesting exploration of pipeline approach to identify claims on social media, assess their legitimacy and virality but approach does not have very high accuracy when it comes to identifying claims and labelling them as misinformation. It also relies on the user taking the initiative to run a Tweet through the pipeline to determine its legitimacy.**

**Rating:** 7
**Confidence:** 3

**Review:**

Overview
- Purpose: To estimate the extent to which misinformation has influenced the course of the COVID-19 pandemic using natural language processing models
- Developed a "misinformation detection" pipeline that is comprised of three components:
1) a claim detector - identifying a social media post as a 'claim'
- ClaimBuster
2) a misinformation classifier - defining a claim as misinformation
- BERT transformer-based machine learning model - Covid-Twitter-Bert-v2
3) a virality measurement - assesses how widespread a false claim is
- retweetability and reach - threshold score of 7.294 based on classification model corresponds to a 'viral' tweet - 25 retweets, 50 comments and 100 likes
-tuning using Adam optimizer (also explored SGD with Nesterov Momentum and other Adam variants)

- Data sources used for model training - CMU-MisCov19 (4600 tweets hand-labeled into 17 categories), USC (2.2 billion tweets)

-Input into pipeline is a single Tweet analyzed for truth value

-Accuracy 78%, F1 score 0.72 for ClaimBuster; 84% and 0.81 for misinformation model;

Clarity
-the sensitivity and specificity metrics of the models for claim detection, misinformation classification and virality assessment are not clearly presented - a table with this information would be nice

Originality
- is there a clustering algorithm or approach that can be used to categorize claims instead of hand labelling, which is not feasible with larger datasets like the USC 2.2 million tweet data (even the 4600 tweets from CMU-MisCov19 is way too much to hand label)
- innovation beyond what is currently available is not well described - what is better about your approach than what already exists (i.e., False Information and Intelligence Desk)

Significance
- misinformation is prevalent and heavy social media use contributes to spread of false information; however, this pipeline approach relies on the user taking the initiative to feed a Tweet into the pipeline to evaluate its legitimacy
- approach does not perform well for tweets that are moderately viral
-Twitter/Instagram/YouTube already flag tweets related to COVID-19 vaccination with links to legitimate sources of information and included labels and warning messages on some tweets with disputed or misleading information about COVID-19

---

### Decision · Program_Chairs · 2022-02-19

Winner